# Ketone Body Induction: Insights into Metabolic Disease Management

**DOI:** 10.3390/biomedicines13061484

**Published:** 2025-06-16

**Authors:** Byung Min Yoo, So Ra Kim, Byung-Wan Lee

**Affiliations:** 1Department of Statistics and Data Science, Korea National Open University, Seoul 03087, Republic of Korea; qudalsdb@gmail.com; 2Division of Endocrinology and Metabolism, Department of Internal Medicine, Yonsei University College of Medicine, Seoul 03722, Republic of Korea; mdsrkim12@yuhs.ac

**Keywords:** β-hydroxybutyrate, diabetes, ketogenesis, ketone bodies, metabolic diseases

## Abstract

Ketone bodies (KBs), particularly β-hydroxybutyrate, are crucial metabolites that provide clean and efficient energy, especially during periods of low glucose availability. Ketogenesis is a promising therapeutic avenue for conditions such as obesity, metabolic syndrome, and diabetes. This review aims to summarize the current evidence on ketogenesis across different health conditions and therapeutic modalities, highlighting the potential to mitigate metabolic disorders and diabetes-related complications. By reducing inflammation and oxidative stress, increased KB production provides cardiovascular and neuroprotective benefits. Ketogenesis is enhanced under physiological conditions like pregnancy and fasting, as well as in pathophysiological states such as diabetes and heart failure. Various interventions, including the promotion of endogenous ketogenesis through diet and exercise, drug-induced ketogenesis via sodium-glucose cotransporter 2 inhibitors, and exogenous ketone supplementation, have demonstrated favorable effects on metabolic health. However, challenges remain, including risks such as pathological ketoacidosis and dyslipidemia. In specific populations, such as lean mass hyper-responders, laboratory lipid profiles might reflect the metabolic privilege. This review will assist in the future clarification of individual differences and optimized therapeutic approaches leveraging ketogenesis for the personalized management of metabolic disorders.

## 1. Introduction

Ketone bodies (KBs), particularly β-hydroxybutyrate (β-HB), serve not only as clean and efficient fuel for the brain, heart, and muscles, but also provide neuroprotection, anti-inflammatory effects, improved metabolic health, and cardiovascular (CV) benefits [1,2,3]. Consisting of acetoacetate, β-HB, and acetone, KBs are primarily produced in the liver mitochondria through the breakdown of free fatty acids (FFAs) and used peripherally as an alternative energy source when glucose is scarce [4]. These benefits manifest under various conditions, including natural physiological ketosis during the neonatal period and pregnancy; induced physiological ketosis during prolonged fasting, exercise, and ketogenic diets (KDs); and pathophysiological ketosis during diabetes and heart failure. Among these, uncontrolled diabetes can progress to ketoacidosis, a potentially life-threatening condition.

This review aimed to summarize the current evidence on ketogenesis across different health conditions and therapeutic modalities, highlighting the potential to mitigate metabolic disorders and diabetes-related complications.

## 2. Roles of KBs in Metabolic Health

Beyond serving as alternative energy substrates when glucose is scarce, KBs contribute to resistance against inflammatory and oxidative stress while promoting metabolic homeostasis by reducing reliance on insulin-dependent glucose metabolism [5]. The metabolic actions of KBs modulate systemic fuel selection by attenuating glucose oxidation in the peripheral tissues, exerting feedback inhibition on adipose tissue lipolysis during ketosis, and reducing proteolysis in skeletal muscle [6]. β-HB suppresses NLRP3 inflammasome activation and reduces inflammation independently of its role as a metabolic substrate or canonical signaling pathways, suggesting that β-HB contributes to the anti-inflammatory effects of energy restriction [7]. Ex vivo macrophage studies confirm that elevated β-HB concentrations and reduced insulin levels suppress NLRP3 inflammasome activation [8].

Epidemiological studies have shown that metabolically healthy, normal-weight individuals exhibit higher urinary KB levels compared to metabolically unhealthy, obese populations [9]. Individuals with ketonuria, as opposed to those without, have a lower prevalence of obesity, central obesity, and metabolic syndrome and exhibit better metabolic parameters [10,11]. Moreover, the presence of spontaneous fasting ketonuria is significantly associated with a reduced risk of diabetes and increased weight loss potential, suggesting a protective role against the onset of diabetes [12] and serving as a predictive factor for further weight loss [13]. Therapeutic applications of KDs or exogenous KBs, metabolized into β-HB, show promise for various metabolic disorders, such as obesity and type 2 diabetes (T2D) [14], and provide protection against ischemic injury to the heart, gut, liver, and eyes [15], as well as potential renoprotective effects in diabetic nephropathy [16]. In contrast, transitioning to similar but high-calorie or high-fat diets may worsen these conditions [15].

## 3. Pathways to Ketosis: Endogenous Ketogenesis and Exogenous Ketone Supplementation in Health and Disease

Endogenous ketogenesis is initiated by various physiological and pathophysiological conditions, as well as therapeutic interventions. Physiological factors promoting ketogenesis include fasting, aerobic exercise, and KDs, whereas pathophysiological conditions such as diabetes and heart failure, as well as certain pharmacological interventions—particularly glucose-lowering agents—can also induce ketogenesis. Supplementation with exogenous ketones also complements these approaches, collectively enhancing metabolic resilience (Figure 1).

Figure 2 illustrates the ranges of blood KB concentrations observed across physiological, pathophysiological, and pathological states of ketogenesis, with concentrations primarily reflecting β-HB levels. In natural physiological conditions, KB levels remain low in healthy adults (<0.1 mM) but can rise modestly during pregnancy and infancy (0.2–1.0 mM) due to altered energy demands and hormonal regulation [17]. Induced physiological ketogenesis through fasting, aerobic exercise, and KDs results in moderate elevations of blood KBs (0.5–5.0 mM), with fasting typically producing levels of approximately 1–2 mM in the short term and up to 7.0 mM with prolonged restriction [5,18]. Pathophysiological ketogenesis, seen in chronic metabolic conditions such as T2D and heart failure, is characterized by mildly elevated KB levels (0.1–1.5 mM), often indicating subclinical metabolic stress [19]. In contrast, pathological ketogenesis, as observed in diabetic ketoacidosis (DKA), results in severe hyperketonemia (3.0–25.0 mM), indicating a loss of metabolic control and requiring urgent clinical intervention [5,20].

### 3.1. Endogenous Ketogenesis: Adaptive Functions and Metabolic Implications

#### 3.1.1. Physiological Ketogenesis

With respect to ketogenesis during significant life stages like pregnancy and infancy, natural physiological ketogenesis occurs. This is prompted by increased placental hormone and cytokine levels during the second half of pregnancy, leading to catabolic states that promote lipolysis [21] and thereby increase the availability and utilization of FFAs as maternal energy sources to preserve glucose primarily for fetal needs [22]. During pregnancy, ketogenesis significantly increases, with nocturnal KB production approximately three times greater in pregnant than non-pregnant women [23,24].

Physiological ketogenesis can be induced by fasting, aerobic exercise, and KDs. This is an evolutionary adaptation to food-deprived conditions, allowing the energy substrate to shift from glucose to KBs. During fasting, the production and concentration of KBs increase rapidly, then plateau after approximately 5 days [25,26]. A negative feedback mechanism, mediated by the anti-lipolytic and insulinotropic effects of KBs, limits further production. This stabilizes KB levels slightly above the rate of utilization, resulting in minimal urinary excretion [26].

Compared to strength and strenuous exercise, which primarily rely on glucose as an energy source, aerobic exercise requires continuous oxidation of various substrates, including KBs. During aerobic exercise, KBs are used as a fuel source, and their levels are markedly elevated during the post-exercise recovery period. Exercise-trained skeletal muscle demonstrates an enhanced capacity to utilize KBs as fuels [6]. Blood KB concentrations can increase to as high as to 1.8 mM following exercise, a phenomenon known as post-exercise ketosis [18], and circulating β-HB levels of 1–2 mM have been observed following exhaustive exercise [18,27]. The process of keto-adaptation refers to the physiological transition where the body optimally uses KBs as a primary fuel source during aerobic exercise. The increase in ketogenesis, facilitating keto-adaptation, can result in enhanced aerobic exercise capacity [28]. The health benefits associated with exercise may be partly mediated through elevations in circulating KBs following bouts of aerobic exercise [29].

Although various KD approaches have been investigated for metabolic diseases, their broader clinical utility remains a subject of ongoing debate. In healthy individuals, KDs can reduce body weight more effectively than calorie-restricted or low-fat diets [30,31,32]. However, in a 12-week randomized controlled trial (RCT), transient increases in apolipoprotein B, C-reactive protein, and postprandial glycerol were observed. The authors concluded that a KD may not necessarily confer cardiometabolic benefits proportional to the weight loss achieved [33]. In patients with obesity, the beneficial effects of KDs, including weight loss, decreased blood glucose levels, and improved insulin sensitivity, are generally recognized [31,34]. In individuals with T2D, KDs may exhibit mixed effects on CV risk factors. For example, in a systematic review of 14 studies, KDs induced significant short-term body weight and fat loss (particularly abdominal fat) within 3–8 weeks. However, in long-term studies (≥12 months), differences in body weight between individuals on KDs and control diets were often nonsignificant in the absence of high adherence [35]. In terms of glycemic control, several studies have demonstrated that a KD improves glycemic control in patients with T2D [36,37,38]. Moreover, a systematic review indicated glycated hemoglobin (HbA_1C_) improvement as early as 3 weeks that was sustained for at least 1 year [35]. However, another study demonstrated no differences in the change from baseline at 12 months between KDs and control diets, and evidence of the superiority of KDs over other diets remains limited [39]. In metabolic dysfunction-associated steatotic liver disease (MASLD), KDs exhibited beneficial effects in the short to medium term (2 weeks to 6 months), independent of calorie and fat intake [40]. Furthermore, contrary to the common belief that high-fat intake promotes hepatic steatosis, a normocaloric KD suppressed de novo lipogenesis and enhanced fat oxidation, leading to weight loss and reduced hepatic fat [41].

Given that long-term adherence to a KD can be challenging for many individuals, dietary guidelines now emphasize the importance of tailoring interventions to individual needs [42,43]. For example, with a 1400 kcal KD, exercise is permitted and encouraged, whereas under a very-low-calorie KD (VLCKD) or a very-low-energy KD (VLEKD), which may provide only 7–900 kcal per day, exercise is not recommended.

#### 3.1.2. Pathophysiological Ketogenesis

Pathological conditions can trigger ketogenesis as a compensatory response to maintain homeostasis. In individuals with T2D, where glucose is abundant rather than deficient, the synthesis of KBs appears paradoxical from the standpoint of energy metabolism. However, the range of ketosis varies widely, from mild ketonemia to overt ketoacidosis, with total KB concentrations in diabetes ranging from undetectable to 13.28 mM and positively correlated with FFA levels [44]. This variability in KB levels may manifest clinically, ranging from well-controlled metabolic states to severe decompensation, such as DKA. Initially, in T2D, the body compensates for insulin resistance with hyperinsulinemia to maintain normal blood glucose levels. Over time, as pancreatic insulin secretion frequently diminishes, these compensatory mechanisms fail and cells respond insufficiently or ineffectively to insulin, thereby exacerbating chronic hyperglycemia. When insulin levels are insufficient, the body initiates ketogenesis, utilizing FAs as an alternative energy source—a shift particularly evident during marked insulin deficiency [14]. Individuals with T2D might exhibit the presence of KBs even without significantly elevated HbA_1C_ levels [4]. This phenomenon reflects relative insulin deficiencies or heightened insulin resistance, especially during acute stressors such as illness and infection [4]. Interestingly, the presence of β-HB in diabetes may not always indicate metabolic decompensation. Our prior work reported that higher serum β-HB levels in individuals with metabolically uncompromised T2D are associated with a better response to anti-diabetic treatment, resulting in greater improvement in hyperglycemia compared to those with suppressed or absent ketogenic function [45].

In individuals with heart failure, impaired mitochondrial oxidation of FAs and glucose leads to an energy-deprived myocardium, prompting increased cardiac oxidation of KBs as an alternative energy source [46]. Both the elevated concentration and enhanced availability of KBs, resulting in endogenous ketogenic ketonemia, represent a crucial adaptive mechanism that enhances the metabolic resilience of the heart. This shift toward enhanced cardiac KB utilization in heart failure is driven not only by systemic ketosis but also by intrinsic upregulation of cardiac ketolytic enzymes [47]. KB utilization yields more adenosine triphosphate (ATP) per unit of oxygen than FA oxidation, which may support the energetically compromised hypertrophied heart. Although KB oxidation does not improve cardiac efficiency, it aids cardiac function by supplying additional fuel and bolstering mitochondrial tricarboxylic acid (TCA) cycle activity [46,48,49,50]. The hearts of patients with diabetes show enhanced uptake of total KBs and β-HB compared to non-diabetics, underscoring a metabolic advantage from ketogenesis [51]. Emerging evidence suggests that therapeutic approaches, including intravenous KB administration, sodium-glucose cotransporter 2 (SGLT2) inhibitors, and ketone ester cocktails, may improve cardiac function in heart failure patients [46,52]. Alternate-day KDs—not continuous ones—conserve ketogenesis in the liver, serving as a protective mechanism against heart failure [53]. Additionally, KBs behave as cardioprotective agents with antioxidant and anti-inflammatory properties, influencing metabolism and gene expression through G protein-coupled receptors and histones [54]. The beneficial effects of KBs in heart failure might extend beyond the heart itself, including the modulation of immune responses, reduction of fibrosis, and promotion of angiogenesis and vasodilation [47].

#### 3.1.3. Ketogenesis Induced by Glucose-Lowering Agents

SGLT2 inhibitors are drugs that enhance renal glucose excretion by blocking SGLTs, which are essential for glucose reabsorption in the kidneys. Consequently, they lower blood glucose levels and promote weight loss [55]. Along with decreased insulin secretion resulting from reduced blood glucose levels, SGLT2 inhibitors increase systemic and tissue β-HB levels by upregulating ketogenic enzymes and transporters in the liver, kidneys, and intestine. This suggests integrated physiological changes for KB metabolism associated with SGLT2 inhibitor use [55].

Furthermore, SGLT2 inhibitors improve CV and renal outcomes, with benefits extending to patients without diabetes. This occurs by shifting the metabolic substrate for the heart and kidney toward KBs [56]. SGLT2 inhibitors effectively treat heart failure, likely through beneficial metabolic effects that include KB-mediated improvements in cardiac energy metabolism, ventricular remodeling, and inflammation. These findings underscore the therapeutic potential of KB production [57]. In diabetic kidney disease, SGLT2 inhibitors provide significant renal protection, in part by elevating KBs levels, which may help restore renal ATP production and suppress pathological mammalian target of rapamycin complex 1 hyper-activation, thereby reducing kidney damage and proteinuria. Additionally, they activate the SIRT1/PGC-1α/fibroblast growth factor 21 (FGF21) or nuclear factor erythroid 2-related factor signaling pathways, leading to a reduction in inflammation, oxidative stress, and cellular dysfunction [58,59,60]. Importantly, the effects of β-HB appear to be organ specific. In the heart, β-HB serves as an efficient energy substrate due to the high expression of succinyl-CoA:3-ketoacid CoA transferase (SCOT), whereas in the kidney—where SCOT expression is downregulated—it confers renoprotective effects by reducing inflammation, oxidative stress, and fibrosis [61,62].

Glucagon-like peptide-1 (GLP-1) receptor agonists (GLP-1RAs) are a class of drugs used in the treatment of diabetes and obesity. GLP-1 is an incretin hormone produced by L-cells in the ileum and colon that enhances glucose-dependent insulin secretion, suppresses postprandial glucagon, reduces food intake by promoting satiety, slows gastric emptying, decreases nutrient absorption, and improves lipid metabolism, contributing to glucose-lowering and weight-reducing effects [63,64].

Studies have suggested a potential association between GLP-1RAs and ketoacidosis. However, the evidence is mixed; some studies indicate that GLP-1RAs promote ketoacidosis [65], while others find the opposite [66]. The mechanisms underlying these differences are not yet fully understood but may involve the following factors. While physiological GLP-1 decreases ketogenesis by increasing the insulin-to-glucagon ratio [67,68], GLP-1RAs—by achieving pharmacological levels of GLP-1—exert additional systemic effects, such as appetite suppression [69], which may be detrimental and lead to ketoacidosis. In patients with diabetes, GLP-1RA–induced DKA has been reported at lower rates (8.33 per 1000 or 3.9%) than with SGLT2 inhibitors (12.8%) [70,71], with insulin withdrawal or dose reduction appearing to be a key trigger. Insulin suppresses ketogenesis by inhibiting adipose tissue lipolysis and reducing hepatic cAMP levels, while enhancing the peripheral uptake and utilization of KBs [14]. Deficiency or resistance thereof increases lipolysis and acetyl-CoA availability, promoting ketosis and potentially leading to DKA [72,73]. Consequently, insulin deficiency plays a central role in KB accumulation and the development of ketoacidosis during GLP-1RA therapy in both T1D and T2D [74,75]. Other contributing factors beyond impaired islet function, as reflected by low C-peptide levels, are restricted intake, gastrointestinal adverse events, and dehydration [70]. When GLP-1RAs are used for weight loss in individuals without diabetes, GLP-1RA–associated ketoacidosis is rare and typically presents as euglycemic ketoacidosis [76]. This form is similar to starvation ketoacidosis and likely triggered by reduced intake due to appetite suppression and the gastrointestinal side effects of GLP-1RA or occurring concurrently during GLP-1RA therapy via a KD and alcohol use. This reduces glucose availability, lowers insulin, and elevates glucagon, collectively enhancing lipolysis and stimulating hepatic KB production [77,78]. These cases highlight the need for adequate hydration and carbohydrate intake [76].

Data on whether GLP-1RA therapy induces metabolically beneficial ketonemia and the real-world incidence of ketosis remain limited. Moreover, unlike SGLT2 inhibitors, it is unclear whether GLP-1RA–associated KB production confers pleiotropic metabolic effects. Further investigation is warranted to define the optimal thresholds of reduced carbohydrate intake and insulin dose reduction during GLP-1RA therapy that facilitate metabolically favorable ketosis without increasing the risk of detrimental ketoacidosis.

#### 3.1.4. Pathological Ketogenesis

Pathological ketosis, characterized by abnormally high levels (3.0–25.0 mM) of KBs in the blood, differs from physiological ketosis (0.2–2 mM) and is caused by a variety of endogenous and exogenous factors. Common causes include diabetic or alcoholic ketoacidosis, salicylate intoxication, and rare inborn errors of metabolism typically stemming from insulin deficiency or the activation of toxic metabolic pathways by ethanol or drugs [4]. DKA is an acute pathological condition characterized by hyperglycemia, ketosis, and metabolic acidosis. It primarily results from insulin deficiency, accompanied by elevated counterregulatory hormones such as glucagon, catecholamines, cortisol, and growth hormone [79,80]. This hormonal environment stimulates lipolysis in adipose tissue, releasing large amounts of FFAs into the bloodstream, which act as both substrates and stimulators of hepatic ketogenesis and have impaired hepatic re-esterification. Glucagon-induced reductions in malonyl-CoA alleviate the inhibition of carnitine palmitoyltransferase 1, enhancing mitochondrial FFA transport and β-oxidation. The resultant excess acetyl-CoA, along with depleted oxaloacetate, is diverted toward excessive KB production, leading to the release of substantial quantities of β-HB and acetoacetate into the blood [81,82]. These KBs are strong organic acids that fully dissociate at physiological pH, generating a significant hydrogen ion load that overwhelms the buffering capacity of serum bicarbonate, resulting in a high anion gap metabolic acidosis. In DKA, the rate of KB production consistently exceeds the combined rates of KB utilization and excretion [4].

Although the incidence of DKA in the overall T2D population is low [83], the proportion of T2D among individuals with new-onset diabetes presenting with DKA—known as ketosis-prone T2D—is as high as 35% [84] to 42% [85]. The clinical course following DKA differs between new-onset ketosis-prone T2D and previously diagnosed T2D [84]. While both groups show markedly impaired insulin secretion at DKA onset, only new-onset ketosis-prone T2D subjects demonstrate significant β-cell recovery and insulin independence during follow-up. In contrast, most previously diagnosed T2D cases required ongoing insulin [86]. These findings suggest that new-onset ketosis-prone T2D may represent a distinct T2D subtype with a favorable trajectory despite severe initial presentation [87,88]. Further studies are needed to clarify the mechanisms underlying DKA development and the favorable clinical course in patients with ketosis-prone T2D.

### 3.2. Exogenous Ketone Supplementation

Consuming ketone monoester (KME) supplements rapidly increases circulating KB concentrations, typically exceeding those achieved by adhering to a KD [89]. This provides a practical, non-drug strategy to achieve nutritional ketosis without drastically reducing carbohydrate intake, which may be beneficial for individuals at increased cardiometabolic risk. In adults with obesity, a 14-day supplementation of KME supplementation before meals enhances glucose regulation, improves vascular function, and lowers cellular inflammation [90]. Furthermore, in people with T2D, oral ketone supplements may offer therapeutic effects by improving metabolic control, reducing inflammation and oxidative stress, and enhancing CV function, thereby interrupting the vicious cycle underlying T2D pathophysiology and associated CV risk [91].

Regarding exercise performance, ketone supplementation has been theoretically proposed to serve as an alternative energy source—helping conserve carbohydrate stores, enhance post-exercise glycogen replenishment, reduce muscle protein breakdown, and act as a metabolic regulator and signaling molecule [92]. However, rapid elevations in KB concentrations with KME supplementation may inhibit FA mobilization during aerobic exercise through negative feedback, and supplement-induced ketosis appears to have minimal effects on muscle glycogen preservation or carbohydrate oxidation during endurance activity [89]. Furthermore, co-ingestion of KME and carbohydrates does not alter exogenous and plasma glucose oxidation, nor does it affect the metabolic clearance rate of glucose during exercise in men compared with carbohydrate alone [93]. Although oral KME supplementation increases plasma β-HB concentrations, it does not prevent a decline in muscle force or modify plasma inflammatory cytokine responses following eccentric exercise [93]. A meta-analysis further concluded that ketone supplementation has no consistent effect on exercise performance—including sprinting and events lasting up to 50 min—or on metabolic, respiratory, CV, or perceptual responses during physical activity [94].

## 4. Navigating the Challenges of Ketogenic Strategies: Balancing Health Risks and Therapeutic Benefits in the Management of Metabolic Disorders

This section aims to explore various concerns—particularly with regard to lipid profile changes and CV risks—associated with ketogenic conditions induced by factors such as nutrition and exercise, as well as the controversial situation of individuals who are lean mass hyper-responders (LMHRs).

### 4.1. Ketogenesis Induced Lipid Profile Changes and Cardiovascular Risk

Under ketogenic conditions—whether induced by physiological states such as fasting or exercise or through dietary modifications—the body’s metabolism broadly shifts toward increased fat utilization. However, the lipid profile associated with the ketogenic state has not been consistently defined and varies among studies.

Fasting, including intermittent fasting, influences lipid metabolism through multiple mechanisms [95]. In the short term, fasting lowers insulin levels, increases hormone-sensitive lipase activity, and promotes the release of FFAs, which are subsequently converted into KBs by the liver. Initially, the triglyceride (TG) levels in very-low-density lipoproteins (VLDLs) may transiently rise; however, they typically decrease after 24–48 h as ketogenesis becomes dominant and peripheral tissues begin to utilize FFAs directly. Low-density lipoprotein cholesterol (LDL-C) may temporarily increase due to hemoconcentration or cholesterol efflux from adipocytes but generally declines with sustained fasting and weight loss. Different forms of intermittent fasting, such as alternate-day fasting and time-restricted feeding, improve lipid profiles by reducing LDL-C and TG levels while increasing high-density lipoprotein cholesterol (HDL-C) levels, particularly in individuals with elevated baseline lipid levels [96]. Alternate-day fasting promotes weight loss and ketosis, improves LDL-C particle size, and reduces the proportion of small, dense LDL-C particles [97]. However, increases in LDL-C have been reported, potentially due to dietary changes on feeding days. Fasting physiologically activates PPARα and induces FGF21, thereby enhancing fat oxidation, TG clearance, and ketogenesis [98]. KBs, such as β-HB, also regulate lipolysis through activation of G protein-coupled receptor 109A, also known as hydroxycarboxylic acid receptor 2, thereby helping to stabilize lipid levels and prevent excessive FFA release. Overall, intermittent fasting appears cardioprotective when paired with healthy eating on feeding days [99,100].

Aerobic exercise has modest LDL-C-lowering effects or can be neutral but consistently raises HDL-C levels and lowers TG levels [101]. Numerous studies and meta-analyses have confirmed that aerobic exercise, even without dietary change, leads to a modest yet significant increase in HDL-C. The reduction in TG is mediated by increased lipoprotein lipase activity in skeletal muscle, which enhances the clearance of TG from VLDL during and after exercise [102]. Many individuals experience little change in LDL-C from exercise alone unless significant weight or fat loss occurs. While a reduction of approximately 1 mg/dL in LDL-C per kilogram of weight loss through exercise has been reported, exercise may also favorably alter LDL-C particle distribution toward a less atherogenic profile. After months of training, the proportion of small dense LDL-C decreases and average LDL-C particle size increases, even if total LDL-C remains constant.

Although KDs appear to promote a more favorable lipid profile in healthy individuals or those with obesity, they may exert more heterogeneous effects in patients with T2D. For example, in normal-weight individuals, KDs lower TG and total cholesterol levels and increase HDL-C levels [103]. In patients with obesity, KDs also reduce the levels of TGs and LDL-C and increase those of HDL-C, alongside promoting weight loss [32,104]. However, in a systematic review of 14 studies involving patients with T2D, the effect of KDs on lipid profiles appeared inconsistent, with some studies reporting improvements in LDL-C and TG levels while others observed no significant changes or even elevations in TGs [35]. Another systematic review and meta-analysis of RCTs [39] indicated minimal changes in HDL-C and no effect on LDL-C, suggesting that KDs do not worsen lipid profiles compared to high-carbohydrate diets. Variability in lipid responses to KDs highlights the need for personalized monitoring.

The impact of a KD on overall and cardiovascular disease (CVD) mortality remains inconclusive owing to the paucity of available studies. Reducing carbohydrate intake and enhancing insulin sensitivity may confer beneficial effects on CVD risk factors. Conversely, the high-fat content of the KD, particularly saturated fat, may exert detrimental effects on lipid metabolism, trigger inflammatory responses, and enhance CVD risk [105]. In studies of patients with T2D that assessed indirect markers rather than actual CV events, very high adherence to a KD for up to 2 years was demonstrated to reverse metabolic, inflammatory, and dysglycemic biomarkers, as well as reduce the estimated 10-year atherosclerotic CV risk [34]. Two representative studies analyzing the association between KDs and mortality or CV outcomes in large general populations reported conflicting results. Based on National Health and Nutrition Examination Survey data, adherence to a KD in adults in the U.S. was associated with reduced all-cause mortality without an increased risk of CVD-related death. This suggests that, despite the high fat intake, the KD does not increase mortality related to CV conditions. However, this study relied exclusively on single-time, self-reported dietary questionnaires, potentially limiting the reliability of long-term dietary pattern estimation [105]. In a prospective cohort from the UK Biobank, participants who completed at least one 24 h dietary questionnaire and adhered to a low-carbohydrate high-fat (LCHF) diet, characterized by reduced carbohydrate intake that was insufficient to induce sustained nutritional ketosis, were analyzed [106]. Over a median follow-up of 11.8 years, individuals on LCHF diets had significantly higher LDL-C and apolipoprotein B levels, as well as a higher incidence of major adverse CV events (9.8% vs. 4.3%, respectively), than those following standard diets. Moreover, fat from animal sources was positively associated with increased LDL-C and apolipoprotein B levels. However, UK Biobank participants are generally healthy and exhibit lower baseline LDL-C, possibly underestimating the major adverse CV event risk. To date, major guidelines advise caution regarding the KD in T2D, citing limited glycemic benefit, increased LDL-C levels, and concerns about hypoglycemia, ketoacidosis, nutrient deficiencies, poor adherence, and uncertain long-term safety, with recommendations for professional support if a very-low-carbohydrate intake is adopted [107,108].

Interpreting the benefits or risks of KDs requires caution owing to confounding factors such as medication use, physical activity, and dietary adherence. Many studies are short-term, with high dropout rates and inconsistent macronutrient composition, often failing to confirm sustained ketosis. Future research should involve longer-term (≥24 months), well-controlled trials that carefully account for energy intake, nutrient quality, and lifestyle factors to elucidate the true impact of KDs.

### 4.2. LMHR

LMHRs are a subpopulation characterized by lean body mass index (BMI) <25 kg/m^2^, LDL-C levels ≥200 mg/dL, HDL-C levels ≥80 mg/dL, and TG levels ≤70 mg/dL following a KD [109]. A low BMI is strongly associated with large increases in LDL-C when undertaking a carbohydrate-restricted diet, whereas moderate carbohydrate reintroduction ameliorates elevated LDL-C levels on a KD [109,110,111]. The lipid energy model (LEM) explains this metabolic process by proposing that lean individuals on a carbohydrate-restricted diet increasingly rely on fat as an energy source [112]. This metabolic shift enhances the hepatic production and peripheral absorption of TGs carried by VLDL, mediated by lipoprotein lipase. Consequently, this metabolic adaptation causes significant elevations in LDL-C and HDL-C levels, along with lower TG concentrations.

Multiple genes are believed to affect lipoprotein lipase, LDL-receptor function, VLDL and LDL clearance, and lipoprotein remodeling. Combined with a high dietary intake of saturated fat and cholesterol, these genetic factors can lead to unpredictably high LDL-C levels [113]. The intestinal microbiota of LMHRs converts more cholesterol after the introduction of a KD, resulting in increased LDL-C levels [114]. In lean, healthy women undergoing carbohydrate restriction, body composition and energy metabolism markers, rather than saturated fat, are the major drivers of LDL-C levels, consistent with the LEM [115]. Although some studies suggest that increased LDL-C levels in LMHRs may not be associated with an increased risk of atherosclerotic CVD [116,117], this remains a subject of debate. Therefore, the concept of a “metabolically privileged” phenotype should be viewed as a hypothesis, supported by mechanistic insights such as the LEM but lacking definitive long-term outcome data. It is also hypothesized that increased ketogenesis in LMHRs may play a protective role against metabolic syndrome and diabetes (Figure 3).

## 5. Conclusions

This review provides an integrated overview of the strategies for increasing β-HB levels and presents a conceptual framework that positions KBs along a continuum from physiological adaptation to pathological states. The exploration of KBs reveals their complex roles both as metabolic fuels and in therapeutic strategies (Figure 4). These functions are associated with improved cardiometabolic outcomes in conditions such as obesity, metabolic syndrome, and diabetes. Collectively, the beneficial effects of ketogenesis enhance the body’s metabolic resilience. Anti-diabetic medications, such as SGLT2 inhibitors, enhance ketogenesis to varying degrees, providing protective effects against inflammation, oxidative stress, and renal and CV complications. Although elevated LDL-C levels are observed in ketogenic conditions, particularly in the rare subpopulation of LMHRs, the resulting CV risk remains uncertain and warrants further investigation. Future research should more precisely clarify the individual differences and optimize therapeutic approaches leveraging ketogenesis for the personalized management of metabolic disorders.

## Figures and Tables

**Figure 1 biomedicines-13-01484-f001:**
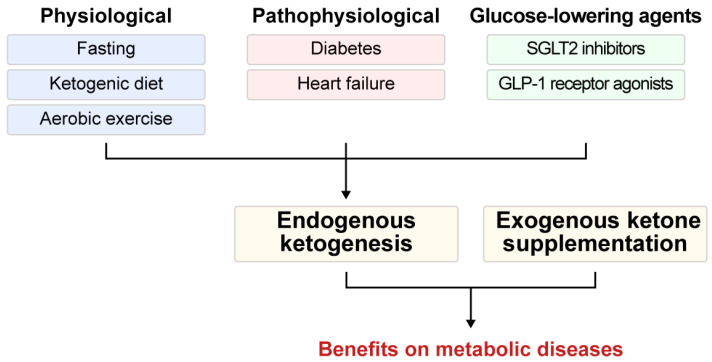
Strategies to induce ketosis. Endogenous ketogenesis can be induced by physiological conditions, including fasting, aerobic exercise, and a ketogenic diet; pathophysiological states, including diabetes and heart failure; and therapeutic interventions, including treatment with glucose-lowering agents. These strategies to promote ketogenesis, along with exogenous ketone supplementation, can enhance metabolic resilience. GLP-1, glucagon-like peptide-1; SGLT2, sodium-glucose cotransporter 2.

**Figure 2 biomedicines-13-01484-f002:**
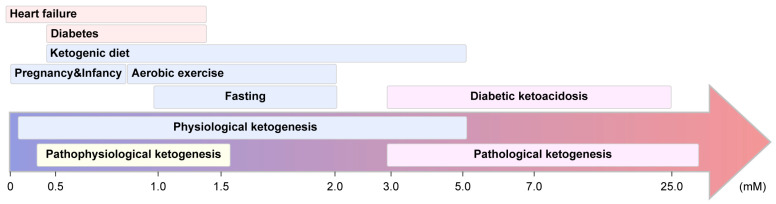
Spectrum of blood ketone body concentrations (mM) and their associated metabolic states. Physiological ketogenesis (0.5–5.0 mM) is induced by fasting, exercise, or a ketogenic diet. Pathophysiological ketogenesis (0.1–1.5 mM) represents a transitional zone that may reflect either adaptive or dysregulated metabolism. Pathological ketogenesis (3.0–25.0 mM), as observed in diabetic ketoacidosis, indicates a loss of homeostatic regulation and requires clinical intervention.

**Figure 3 biomedicines-13-01484-f003:**
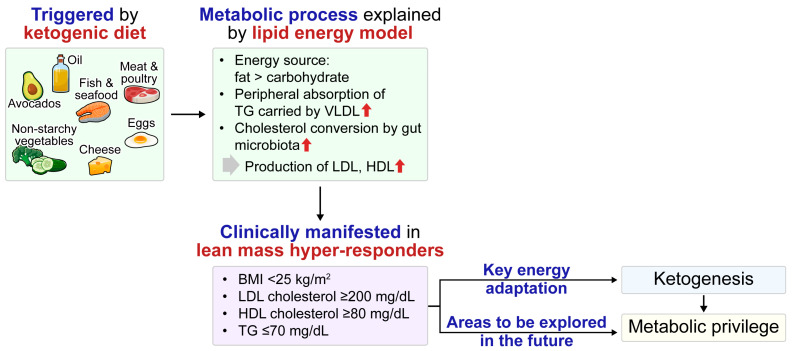
Metabolic adaptations associated with ketogenesis in lean mass hyper-responders illustrated by the lipid energy model, which describes a physiological transition from carbohydrate-based to fat-based energy metabolism. In individuals with a BMI below 25 kg/m^2^ who follow a ketogenic diet, there is evidence of increased peripheral uptake of TGs carried by VLDL and enhanced microbial-mediated cholesterol transformation in the gut. This contributes to elevated levels of both LDL and HDL. Ketogenesis in this population represents a key energy adaptation mechanism, and its association with a metabolically privileged phenotype, despite apparent dyslipidemia, warrants further investigation. The upward red arrow denotes an increase. BMI, body mass index; HDL, high-density lipoprotein; LDL, low-density lipoprotein; TG, triglyceride; VLDL, very-low-density lipoprotein.

**Figure 4 biomedicines-13-01484-f004:**
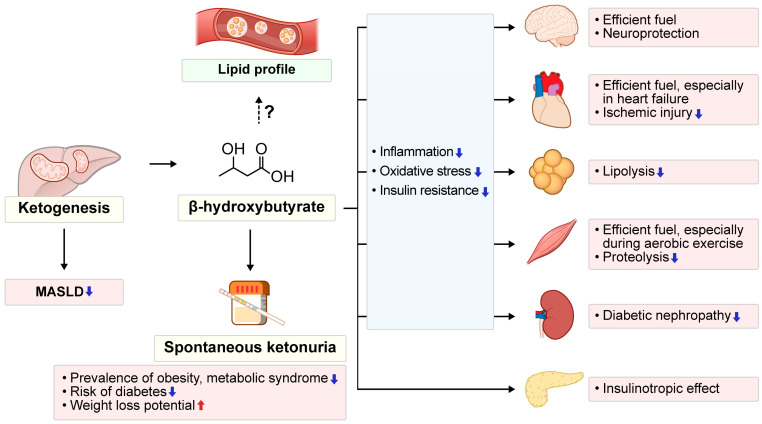
Physiological and metabolic effects of β-hydroxybutyrate. β-Hydroxybutyrate exerts multiple beneficial effects, including reductions in inflammation, oxidative stress, and insulin resistance. It serves as an efficient fuel across various organs—enhancing neuroprotection, supporting the energy demands of cardiac and skeletal muscle, and reducing proteolysis and lipolysis. Furthermore, β-hydroxybutyrate offers protective effects in diabetic nephropathy and exerts insulinotropic effects on pancreatic β-cells. While ketogenesis may reduce MASLD, its impact on the lipid profile remains inconclusive. The upward red arrow denotes an increase, and the downward blue arrow denotes a decrease. The dashed arrow indicates a pathway whose effect remains uncertain. MASLD, metabolic dysfunction-associated steatotic liver disease.

## Data Availability

Data sharing is not applicable to this article as no new data were created or analyzed.

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
