# Peer review of "Ketone Body Induction: Insights into Metabolic Disease Management"

_biomedicines, 2025, doi:10.3390/biomedicines13061484_

Round 1
Reviewer 1 Report
Comments and Suggestions for Authors
The review article by Yoo et al. offers a thorough examination of ketone bodies and their potential roles in managing metabolic diseases. It discusses ketogenesis across a range of physiological and pathological settings and assesses various strategies—dietary, pharmacological, and exogenous—for inducing ketosis. The topic is intriguing, given the increasing interest in ketogenic therapies for conditions such as obesity and diabetes. The authors present a broad collection of evidence, including recent insights into the effects of SGLT2 inhibitors, GLP-1 receptor agonists, and exogenous ketone supplementation. Nonetheless, I have several important concerns regarding the manuscript’s clarity, accuracy, and objectivity.
- Overall, the manuscript reads well and is enjoyable to follow, but a few sentences are confusing or contain errors and therefore need clarification. For example, in the Introduction, line 45, the phrase “when glucose is scare” should be “scarce.” Similarly, Line 40, the sentence “The last pathological condition that causes ketoacidosis can be life-threatening” is unclear: it seems to imply that heart failure causes ketoacidosis, which is incorrect. It should explicitly state that uncontrolled diabetes causing ketoacidosis is life-threatening. These ambiguities should be corrected for clarity.
- The review effectively highlights numerous potential benefits of ketones, consistent with current scientific literature. However, certain conclusions seem speculative, and in some cases, the conclusion is strong. For example, the discussion on lean mass hyper-responders implies that their elevated LDL-cholesterol levels do not contribute to increased cardiovascular risk, referring to them as having a “metabolically privileged” phenotype (Line 413). This assertion remains controversial, and the authors should state that the long-term effect of consistently high LDL-cholesterol in such individuals is yet to be understood. It would be more appropriate to frame it as a hypothesis, supported by relevant research, such as the Lipid Energy Model, while acknowledging the absence of definitive outcome data.
- The manuscript could be strengthened by adding practical context and highlighting some of the limitations of such interventions. For example, ketogenic diets can be difficult for patients to maintain long-term, and many guidelines emphasize individualized dietary approaches. Mentioning this would provide balance. Also, if recent trials or meta-analyses (2023–2024) on ketogenic interventions in conditions like NAFLD, obesity, or diabetes are available, citing them would enhance the review. A brief comparison to other recent reviews or guideline publications could clarify the unique contribution of this work.
- The authors reference numerous pertinent studies, including a 2024 review on nutritional ketosis. However, to ensure the review remains up-to-date and thorough, they should incorporate all major recent publications through 2025. For instance, if new guidelines from professional societies or results from large clinical trials have emerged, these should be cited. Including the most current evidence will strengthen the review’s relevance and completeness.
The overall quality of the English and grammar is strong, requiring only minor refinement. I noted only a few sentences with grammatical issues that could benefit from a light review and polish.
Reviewer 2 Report
Comments and Suggestions for Authors
The article is well written and finely structured. I miss a figure with ketone bodies' metabolism integrated with their effects on various body systems.
I would also suggest a more nuanced approach to the concluding paragraph on ketogenic diets and cardiovascular risk, taking into account that this is more controversial than the manuscript seems to admit. I would suggest the same approach in Figure 1, where only metabolic benefits are extrapolated.
Reviewer 3 Report
Comments and Suggestions for Authors
I congratulate the authors on this useful paper, which highlights the importance of Ketogenic Therapy in individuals with Obesity, MetS, and T2D.
Here are my suggestions to improve this work:
1-Lines 148-152, this sentence is not useful because The metabolism of rats is completely different from humans. In adult humans brown adipose tissue is very limited and a browning effect is not possible. So, Could you consider deleting this sentence or inserting the clarifications?
2- Line 164, regarding exercise under KDs, I would precise that during a classical KD for example 1400 kcal KD, exercise is permitted and encouraged; while under VLCKD or VLEKD where kcals are about 7-900 kcal/die exercise is not allowed.
Please precise this concept in your sentences.
Round 2
Reviewer 1 Report
Comments and Suggestions for Authors
I thank the authors for addressing my questions and concerns.